# Manuscript submission systems and metadata completeness in Crossref: Patterns and associations

Hans de Jonge[ID][1][⊛]*, Bianca Kramer[ID][2][⊛]

1 Dutch Research Council NWO, The Hague, The Netherlands, 2 Sesame Open Science, Barcelona Declaration on Open Research Information, Houten, The Netherlands

⊛ These authors contributed equally to this work.
* h.dejonge@nwo.nl

## Abstract

The importance of open research information, particularly publication metadata, is widely recognised. Crossref is one of the most important infrastructures for registering open metadata as part of DOI record registration. It is widely known, however, that the metadata of many publications is far from complete, with many publishers making certain metadata openly available, but failing to do so for other metadata elements. Publishers' ability to register this metadata with Crossref depends on their capacity to capture and retain this data in their production workflows. Manuscript submission systems are an important, yet largely overlooked, factor in the extent to which publishers make metadata available through Crossref. In this paper, we present the results of an analysis investigating the relation between the level of metadata that publishers deposit with Crossref and the submission systems that they deploy for their journals. We have looked at the 153 publishers with the largest amounts of publications in Crossref and concentrate on the four most commonly used systems: Editorial Manager, ScholarOne, Open Journal Systems (OJS) and eJournalPress. We show that some submission systems appear better suited to capturing certain metadata elements. However, there are always cases where publishers using the same system differ widely in the level of metadata they register, suggesting that technology is not the only prohibiting factor and other considerations are at play.

## 1. Introduction

The rapid adoption of initiatives such as the Barcelona Declaration on Open Research Information (2024), which advocates for open and equitable access to research information, shows that having bibliographic metadata openly available is gaining wide acceptance [1]. Open metadata, especially publication data enables more transparent, reproducible, and inclusive bibliometric analyses. But open research information is also crucial for the more responsible use of bibliometric

**Data availability statement:** The data and code underlying the results presented in the study are available from Zenodo: https://doi.org/10.5281/zenodo.17229316 - The aggregated metadata coverage data from Crossref for 153 publishers across six metadata elements (reused from Van Eck & Waltman); - The manuscript submission systems these publishers use, as collected by the authors, including the validation of this information from the publisher survey; - Questionnaire used to validate the manually collected information on submission system use; - The R scripts used to produce the raincloud plots.

**Funding:** The author(s) received no specific funding for this work.

**Competing interests:** BK is executive director of the Barcelona Declaration on Open Research Information. HdJ is co-chairing the working group on funding metadata under the Barcelona Declaration. The authors write in a personal capacity and views they share in this article do not necessarily express the opinion of their employers.

indicators in research assessment, as promoted by initiatives such as the San Francisco Declaration on Research Assessment DORA (2013) and the Coalition for Advancing Research Assessment (CoARA) [2–3].

Crossref is one of the main sources for open bibliographic metadata [4]. Crossref asks its members to deposit as much rich metadata as possible. When registering a DOI for their publications, publishers can deposit a wide range of metadata elements, from ORCIDs, affiliation information and reference lists to abstracts, funding and license information. Crossref is also an important source for downstream open bibliographic data services like OpenAlex and Lens.

It is well-known from earlier research [5–6] that there is a large variety in the extent to which publishers submit metadata to Crossref. Many publishers do a good job in making certain metadata elements openly available, but fail to do so for other metadata elements. The result is a very patchy landscape with many publications in Crossref lacking crucial information on abstracts, affiliations, licenses or funding information.

In recent years there have been a number of initiatives to improve the quality and coverage of specific metadata elements in Crossref. The initiative for Open Citations (I4OC) has been successful in ensuring that reference data is made available in Crossref. Many of the large and medium-sized publishers submit these data to Crossref, leading to the situation that in 2024 59,8% of publications have an openly available reference list. Similarly, the initiative for Open Abstracts (I4OA) has resulted in 52,3% of publications in Crossref having open abstracts. Three of five of the largest publishers do not support the initiative. This includes Springer Nature, despite a 2022 editorial in its flagship journal Nature calling publishers to make "depositing all relevant metadata on Crossref (...) the norm in scholarly publishing" [7]. For funding information the situation is markedly worse: in 2024 only 24,7% of records in Crossref contain some kind of funding metadata. But as we have shown before [8] many publishers do not submit this data at all or only for a subset of their journals.

Little has been written about the reasons for publishers not to register certain metadata with Crossref. Van Eck and Waltman [6] list three possibilities. Publishers may not be aware of the possibilities, they may not have the expertise or resources needed for it or they may not consider it a priority. It has also been suggested that in some cases publishers do not want to openly share their metadata because they want to monetize this data. Recent steps taken by Elsevier and Springer Nature to have open abstracts removed from OpenAlex have been linked to the potential of using abstracts for AI development and decisions to sell access to abstracts to third party developers [9].

As we have shown before [10] technical capacities of publishers play an important role in their ability to submit metadata to Crossref. In their production workflows publishers operate a range of systems from manuscript handling systems to production and hosting platforms. Publishers' ability to register metadata with Crossref depends on their capacity to capture and retain this data through these systems. At least in the case of funding metadata publishers have signaled problems with the way submission systems support the collection of this metadata [10]. This paper starts

from the hypothesis that this holds true for other metadata as well and that the submission and production systems used by publishers are an important enabling factor in their ability to submit metadata to Crossref.

To test this hypothesis, in this paper we will present the results of an analysis investigating the association between the level of metadata that publishers submit to Crossref and the submission systems that they deploy for their journals. Building on openly shared data from the earlier analysis by Van Eck & Waltman [6] we have looked at the metadata submitted to Crossref by the 153 publishers with the largest publication volume in 2023 and 2024 and concentrate on the four most commonly used submission systems: Editorial Manager, ScholarOne, Open Journal Systems (OJS), and eJournalPress.

We will show that some submission systems appear better suited to capturing certain metadata elements. However, there are always cases where publishers using the same system differ widely in the level of metadata they register, suggesting that technology is not the only prohibiting factor and that other considerations play a role.

Understanding the relationship between submission systems and metadata completeness in Crossref has important practical implications. Publishers can make more informed technology choices and identify best practices from peers using the same systems more effectively. This research can also inform dialogue between publishers and system vendors about improving metadata capture capabilities. By identifying where submission systems enable or constrain metadata capture, we provide an evidence base for coordinated action across the scholarly communication ecosystem.

## 2. Manuscript submission and hosting systems

The literature on digital infrastructure publishers use in their production workflows is limited. This section combines existing research on these systems with information that can be found online about the different systems. Publishers operate a range of third-party systems in their production workflows, from author submission, peer review processing, to production, typesetting and hosting platforms. Typically, a distinction is made between systems for managing manuscripts and peer reviews, and hosting platforms [11]. Manuscript submission systems offer features such as manuscript submission and tracking, reviewer selection, and peer review facilitation, while hosting platforms provide essential workflow tools such as content publishing, access management, and authentication.

Popular proprietary submission systems include Editorial Manager, which is owned by Elsevier [12]; ScholarOne Manuscripts, which was recently acquired by Silverchair [13]; and eJournalPress, which is currently owned by Wiley [14]. The most widely used open-source journal system is Open Journal Systems, which was developed by the Public Knowledge Project (PKP) and is used by almost 50,000 journals worldwide [15–16].

Examples of commercial hosting platforms include Atypon, which was acquired by Wiley in 2016 [17]; HighWire Press, which offers hosting as well as editorial services [18]; and Silverchair [19].

These systems play a vital role in capturing and retaining metadata. Manuscript submission systems represent the starting point of the metadata lifecycle by capturing the initial metadata created by authors, such as keywords, affiliations, and funding information. Integrating these systems with third-party services such as ORCID, ROR, and Crossref's funder registry is important for further enrichment of this initial metadata. Hosting platforms serve a dual role. Firstly, these platforms display the bibliographic metadata of the research they publish. However, they are also often responsible for registering metadata with Crossref on behalf of publishers. This means they are responsible for a significant proportion of Crossref's metadata records [20]. This also shows why excellent integration between the two types of systems is so important for preserving metadata in the journal production workflow.

Although the pivotal role of manuscript submission and production systems for metadata quality in Crossref is acknowledged [11], we know of no research on the relationship between the use of those systems and the degree of completeness of metadata in Crossref. Is there a relation between the extent to which publishers submit certain metadata types to Crossref and the systems they use? Are some systems better at capturing certain metadata elements than others? What patterns can be discerned regarding the levels for metadata elements and the systems publishers use?

To begin to answer these questions, in this paper we present a descriptive analysis of the relationship between the submission system that publishers deploy and the levels of metadata they submit to Crossref. We focus on the submission systems that publishers use as this information is often publicly available whereas information about the use of production and hosting platforms is much harder to obtain.

## 3. Method

### 3.1 Publishers

For this analysis, we use the list of 153 publishers with at least 5,000 journal articles in 2023 and 2024, listed in Van Eck and Waltman [6]. Publishers are identified by Crossref member ID (thus harmonizing variants in publisher names).

For determining the number of journal articles by publisher, Van Eck and Waltman used the publication type 'journal article' in Crossref. This covers not only research articles, but also other types of documents published in journals, such as letters, editorials, book reviews, and corrections. For publication year, print year was used when provided in Crossref, otherwise, another year field was used, typically the year of online publication.

The complete list of publishers was taken from the dataset accompanying Van Eck and Waltman, available on Zenodo [21]. In total, 7,6M journal articles were analyzed, representing 68% of the total number of 11,2M journal articles from 2023–2024 identified in Crossref [21].

### 3.2 Submission systems

For each publisher included in this analysis, an attempt was made to identify the submission system used by the publisher. The initial procedure entailed a manual search of the publisher's website for the name of their submission system. In instances where this information was not readily available on the website, a random spot check was performed on journal level by accessing the submission portal of a particular journal.

To validate the results obtained manually, a short questionnaire was disseminated to all 153 publishers, asking for information regarding the submission and production/hosting systems they use. In instances where publishers employ a combination of systems, they were requested to specify the proportion of journals for which they utilize each system. The survey was disseminated by Crossref via the metadata contact addresses of their members. Formal ethics approval was not sought for this questionnaire as it involved a voluntary survey collecting non-sensitive organizational information from professional contacts acting in their institutional capacity. Respondents were informed of the research purpose at the beginning of the questionnaire [22].

A total of 41 responses were received, meaning that not all manually collected information could be validated. Table 1 provides an overview of the results that were obtained.

For 49 publishers, it proved impossible to identify the system in use. No information could be found on the website, and no response was received on the questionnaire. Consequently, for 104 of the major publishers, information regarding the submission systems they use are available.

**Table 1. Overview of the submission systems deployed by publishers.**

| Submission system | Number of publishers |
|---|---|
| Editorial Manager | 14 |
| ScholarOne Manuscripts | 23 |
| Open Journal Systems (OJS) | 32 |
| eJournalPress | 9 |
| Other, own or no system | 26 |
| Unknown | 49 |
| **Total** | **153** |

While the majority of publishers employ only four providers of submission systems, 23 of the publishers in our dataset operate an alternative system or have developed their own. This includes publishers such as F1000, which utilizes its F1000 platform, MPDI, which employs its submission system Susy [23], EDP Sciences, which uses its system Nestor [24] and the American Chemical Society (ACS) which has developed ACS Paragon Plus. Also included in this category are publishers that do not have a dedicated submission system in place but seem to operate a web form or request submissions to be sent by email.

16 publishers indicated that they operated multiple submission systems alongside each other for different journals. Together these publishers ran around 7,500 journals on these systems. In such instances, publishers were assigned to the submission system that covered the majority of their journal portfolio, as reported in the questionnaire. Cambridge University Press, for instance, utilizes ScholarOne, Editorial Manager, Open Journals System (OJS), Editorial Express and a system called EditFlow. However, given that 370 out of 420 journals are managed through ScholarOne, Cambridge University Press is classified under ScholarOne. Similarly, IOP employs both ScholarOne, Editorial Manager, and eJournalpress; however, the publisher is classified as ScholarOne given that it has indicated that 80% of journals are operated by that system.

Some publishers have indicated that they have recently migrated all their journals to a new submission platform or are in the process of doing so. Wiley, for instance, has developed a new submission system, named Research Exchange Submission [25]. It has announced [26] that as of July 2025 1,000 journals are hosted on the new platform. Because our objective is to ascertain the influence of the systems on metadata registration in 2023/2024, we made the decision to categorize publishers according to the system employed during those years. For Wiley this means we categorized them as using ScholarOne.

Springer Nature constitutes a unique exception. The company has developed a new system, the Springer Nature Article Processing Platform (SNAPP) in 2019 and is gradually moving all of its journals to this platform. In 2023 it announced that over a third of its own journals and nearly half of its OA journals had been moved [27]. In the 2023–2024 period, however, Springer Nature also employed both Editorial Manager and ScholarOne. As we have been unable to obtain information regarding the proportion of journals covered by either of these systems, we have decided to exclude Springer Nature from the data collection.

**Submission systems in the metadata workflow**

Submission systems represent an important component in the journal production workflow that captures metadata. Fig 1 provides a schematic representation of the metadata pipeline from submission to publication and inclusion in Crossref (and other bibliographic systems), also indicating potential bottlenecks in the collection and retention of metadata.

As Fig 1 illustrates, submission systems represent the starting point where initial metadata is captured from authors. However, the complete metadata workflow involves multiple stages and systems. Hosting and production systems are also important in preserving this metadata and ensuring it propagates through the production process until its submission to Crossref. Publishers also work with third-party providers to extract metadata from the manuscript text, for example to extract affiliation and funding information [28] from the acknowledgement section. Finally, deliberate policies of publishers can play a role, where publishers may decide whether to make specific metadata types available through Crossref or not. Each step in this workflow presents potential bottlenecks for the correct capture, retention and propagation of metadata from submission to deposition in Crossref.

In this analysis, the focus is limited to submission systems, as information about production and hosting systems proved much more difficult to obtain (see section 3.5 for a discussion of study limitations) and submission systems play an important role as the first point of metadata capture in the production workflow.

### 3.3 Types of metadata

For this analysis, we looked at the six key types of metadata that were analysed and discussed in van Eck and Waltman [6]. They were taken from a broader range of metadata available in Crossref based on their importance for research discoverability, proper attribution, and open science practices:

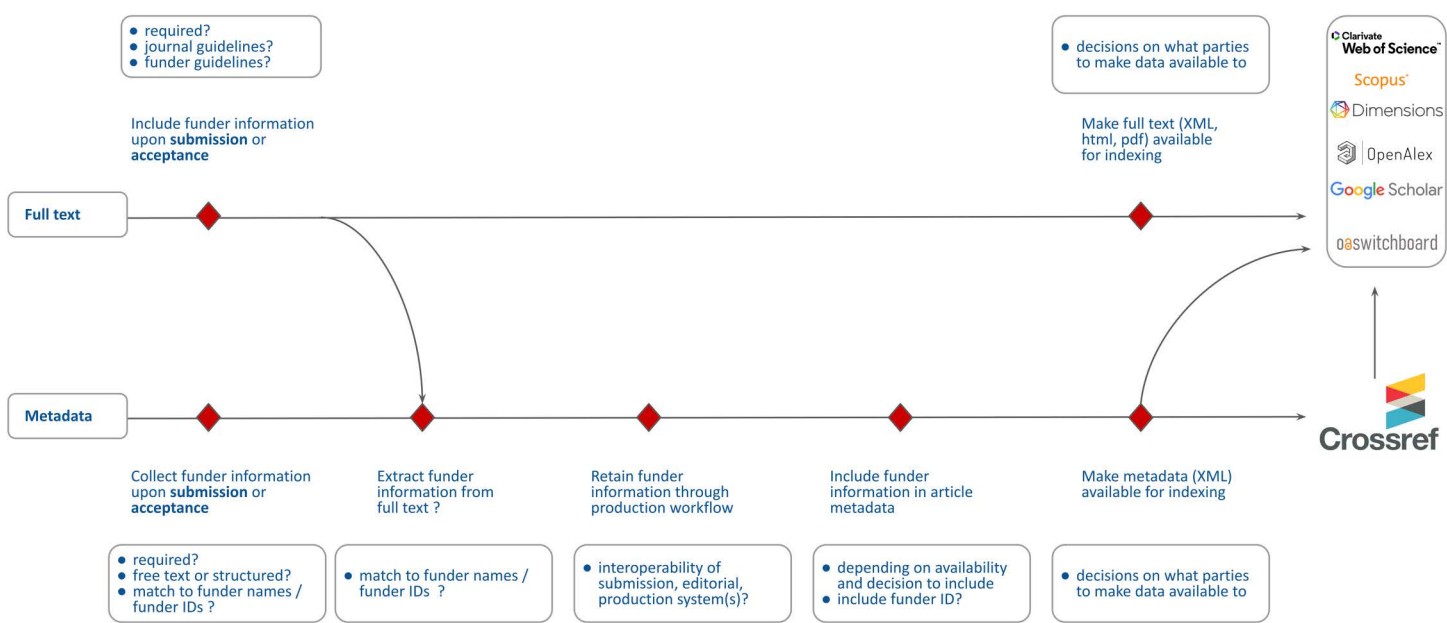

**Fig 1. Schematic representation of metadata pipeline from submission to publication and inclusion in Crossref and example bibliographic systems, with potential bottlenecks indicated by red diamonds.**

- abstracts

- affiliations

- references

- funding information

- ORCIDs

- license information.

For these metadata types, van Eck and Waltman analysed the percentage of Crossref records with type 'journal article' and publication year 2023 or 2024 for which the respective metadata field is populated, for each of the 153 publishers in their dataset [21]. Together, 7,6M Crossref records were analysed for these publishers, representing 68% of all 2023–2024 journal article records in Crossref.

It should be noted that since all journal output is registered as 'journal article' in Crossref, this can also include publication types like editorials, letters, book reviews and other non-article output, though this is generally a minority compared to actual journal articles.

For each metadata type, presence of information in the metadata field was counted as a positive, resulting in a percentage of Crossref records with that metadata type, per publisher. For metadata associated with authors (affiliations and ORCIDs), the information needs to be present for at least one author, but not necessarily for every author.

No further attempt was made to analyse and check the information provided in the metadata fields, e.g., length and textual characteristics of the abstracts, completeness of affiliation information (including presence or absence of ROR), number and completeness of references (including presence or absence of DOIs), type of funding information (funder name and/or ID, award number), authentication of ORCIDs and standardization of license information.

Therefore, the results of van Eck and Waltman, and by extension our study, provide an initial insight into the coverage of metadata (number of records with that metadata type present), rather than their quality.

The percentages metadata coverage per publisher and publication type can be compared to and verified against those in the Crossref Participation Reports (https://www.crossref.org/members/prep/), with the understanding that these cover publications from 2024–2026 compared to 2023–2024 in the original dataset.

### 3.4 Distribution plots

We combined information on submission systems (see 3.2) and metadata coverage (see 3.3.) to plot, for each metadata type, the distribution of metadata coverage (as percentage of Crossref records with that metadata type) per publisher, grouped by submission system. Distribution graphs are shown as combined scatter plots, box plots and violin plots using software for Raincloud plotting developed by Allen et al. [29] and implemented in a Shiny-powered web interface developed by Gabriel Forn-Cuni [30]. In these plots, each dot represents an individual publisher.

### 3.5 Limitations

This study focuses specifically on manuscript submission systems as a factor in metadata completeness in Crossref. We attempted to collect information on production and hosting systems as well, but this proved far more difficult. Publishers rarely disclose which systems they use, they are harder to deduce from publishers' or journals' websites, and it is not possible to directly interact with these systems like it is with submission systems. In addition, the questionnaire responses were too incomplete to provide meaningful insight. This is an important limitation of our study.

As illustrated in Fig 1, submission systems represent only one part of the journal production workflow that captures metadata. Hosting and production systems are just as important in preserving this metadata and ensuring it propagates through the production process until its submission to Crossref. Publishers also work with third-party providers to extract metadata from the manuscript text, for example to extract affiliation and funding information [28] from the acknowledgement section. Finally, deliberate policies of publishers can also play a role, where publishers may decide whether to make specific metadata types available through Crossref or not.

In short: the relationship between submission systems and completeness in Crossref is not straightforward: the chain of systems involved in the production process (from submission to production and hosting) is complex. Metadata can be lost at all stages in that process and therefore it is important to be cautious in our conclusions and avoid claiming direct one-to-one statistical correlations between submission systems and metadata completeness in Crossref.

Another limitation concerns identifying submission systems correctly and completely, especially where publishers may use multiple submission systems across their portfolio of journals. We have made efforts to triangulate our manual detection of submission systems with lists provided by system vendors and self-reporting by publishers through the survey, and documented provenance of detection in the dataset accompanying this study. Nonetheless, for 32% of publishers, the submissions system could not be determined.

While a more detailed study could, in theory, look at the relation between submission systems and metadata coverage at journal level instead of at publisher level, we have chosen not to follow this approach in this study for a number of reasons. First, it would have taken exponentially more time to identify submission systems at journal level. Second, for metadata coverage, we could not have used the existing open dataset by Van Eck and Waltman [21] which is aggregated at publisher level. Third, and most important, our research question explicitly focuses on publishers and their role in making metadata openly available through Crossref. Publishers, not individual journals, make the organizational and policy decisions about metadata deposition, negotiate with system vendors, configure their technical infrastructure, and ultimately bear responsibility for metadata completeness in Crossref. Performing the analysis at the journal level would detract from this focus on publisher responsibilities and obscure the organizational factors that drive metadata practices.

Finally (as mentioned in 3.4), our study provides an initial insight into the coverage of metadata (number of records with that metadata type present), rather than their quality, as similar limitations apply to the original dataset on metadata coverage [21] we re-used for this study.

## 4. Results

In this section, we show the distribution of metadata coverage (as percentage of all journal articles published in 2023–2024 per publisher) grouped by metadata type and submission system. Each figure compares the coverage of one particular metadata type (abstracts, affiliations, references, funding information, ORCIDs and licenses) across the 6 classes of submission systems identified (Editorial Manager, Scholar One, OJS, eJournalsPress, 'own/ other' and 'unknown'. Each data point in the scatter plots represents a single publisher, showing their metadata coverage percentage for that particular metadata type and submission system.

### 4.1 Abstracts

Fig 2 shows the distribution of open abstract coverage in Crossref by publisher, categorized by their submission system. In total, out of 11,2M articles from 2023–2024 in Crossref, 52.7% had abstract metadata. Both Editorial Manager and ScholarOne exhibit a wide distribution: some publishers achieve very high abstract coverage, while others remain low. Overall, ScholarOne appears to be associated with somewhat higher levels of coverage. Both OJS and eJournals Press

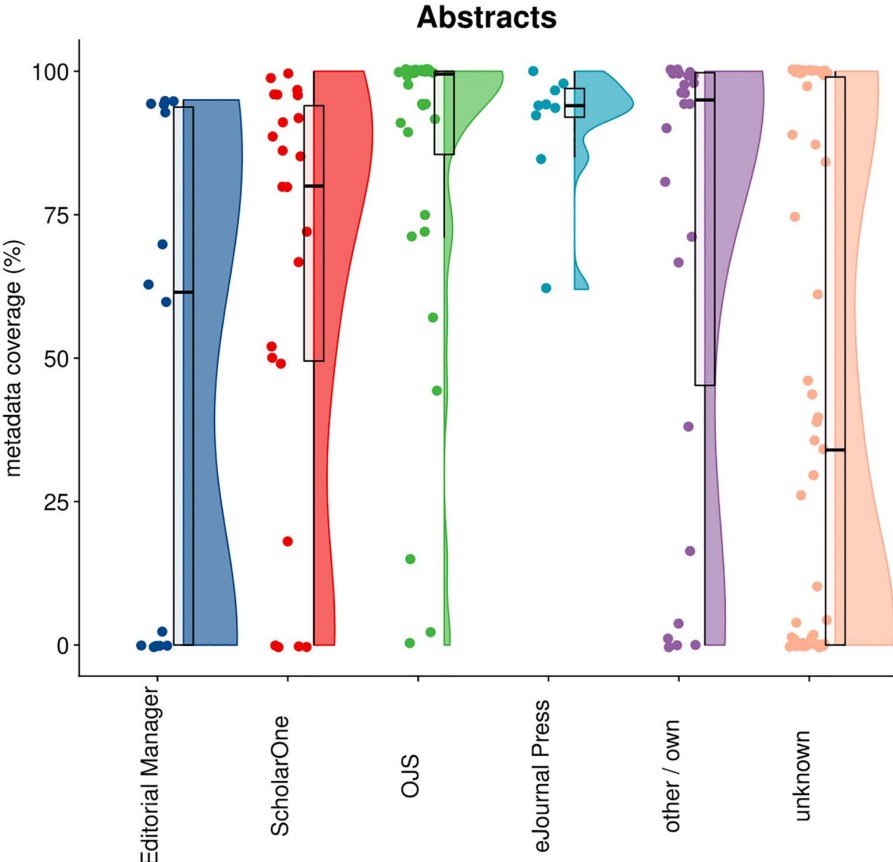

**Fig 2. Distribution of open abstract coverage in Crossref by publisher, categorized by their submission system.**

achieve high overall percentages, although for OJS there are also publishers with much lower coverage. Other submission systems (including in-house solutions) are generally linked to good levels of abstract coverage. For publishers whose submission systems could not be identified, the distribution appears binomial, with most falling into either high or low coverage categories.

When interpreting these findings, several considerations are important. First, not all journal publications will include abstracts, so coverage levels partly reflect the composition of a publisher's journal portfolio. Second, depositing abstracts as metadata in Crossref is not mandatory. The Initiative for Open Abstracts has gained support from many large publishers, but a substantial number of publishers either do not participate or restrict participation to their open access content. In general, the results show that all four submission systems in scope of our analysis are capable of supporting the collection and deposit of abstracts as metadata, but not all publishers make use of this feature.

## 4.2  Affiliations

Fig 3 displays the distribution of affiliation metadata in Crossref by publisher, categorized by submission system. In total, out of 11,2M articles, 30,2% had affiliation metadata.The data shows marked differences in performance across platforms. Both Editorial Manager and ScholarOne seem to generally accommodate the collection of affiliation data quite well, with distributions heavily concentrated at the higher end of the scale (85–100% coverage), though some publishers using these systems still achieve lower coverage rates. Similarly, eJournals Press performs well for this metadata type, with most publishers clustered above 80% coverage. In stark contrast, OJS shows poor affiliation metadata capture, with most publishers not reaching more than 10% coverage and very few exceeding 25%. This pattern confirms known technical limitations in how OJS handles or transmits affiliation data to Crossref in earlier versions of the software [31].

Publishers using "other/own" systems display a bimodal distribution skewed toward lower coverage, while those with "unknown" systems show heavily concentrated low performance, with most publishers below 20% coverage.

As most if not all articles are expected to have affiliation information, and this information can generally be expected to be captured during submission, the success or failure to deposit this information to Crossref could be considered to be mostly reflective of publishers' willingness to make this information available. The strong performance of Editorial Manager, ScholarOne, and eJournals Press suggests these platforms have robust workflows for capturing and transmitting affiliation metadata. Conversely, the consistently poor performance across publishers using OJS points to technical barriers that require system-level improvements. Crossref and PKP recently announced a new partnership that will address (some of these) issues [32].

## 4.3  References

Fig 4 presents the distribution of reference metadata coverage in Crossref by publisher, organized by submission system. In total, out of 11,2M articles, 59,0% had reference metadata.The patterns reveal significant variation in how different platforms handle bibliographic reference data. Editorial Manager shows a broad distribution across the full coverage spectrum, with substantial concentrations both in the mid-range (40–60%) and at higher levels (80–100%), indicating considerable variation in the willingness of publishers to share this data using this system.

ScholarOne shows a similarly wide distribution but with a notable concentration of publishers achieving very high coverage (90–100%), suggesting that many publishers using this platform successfully capture comprehensive reference data. In contrast, OJS demonstrates poor performance, with the vast majority of publishers clustered near zero coverage and very few exceeding 25%.

eJournals Press shows strong performance with most publishers concentrated above 50% coverage, particularly in the 80-100% range. Publishers using "other/own" systems display a binomial pattern, with clusters at both high and low coverage levels, while the "unknown" category shows mixed performance, slightly skewed toward lower coverage.

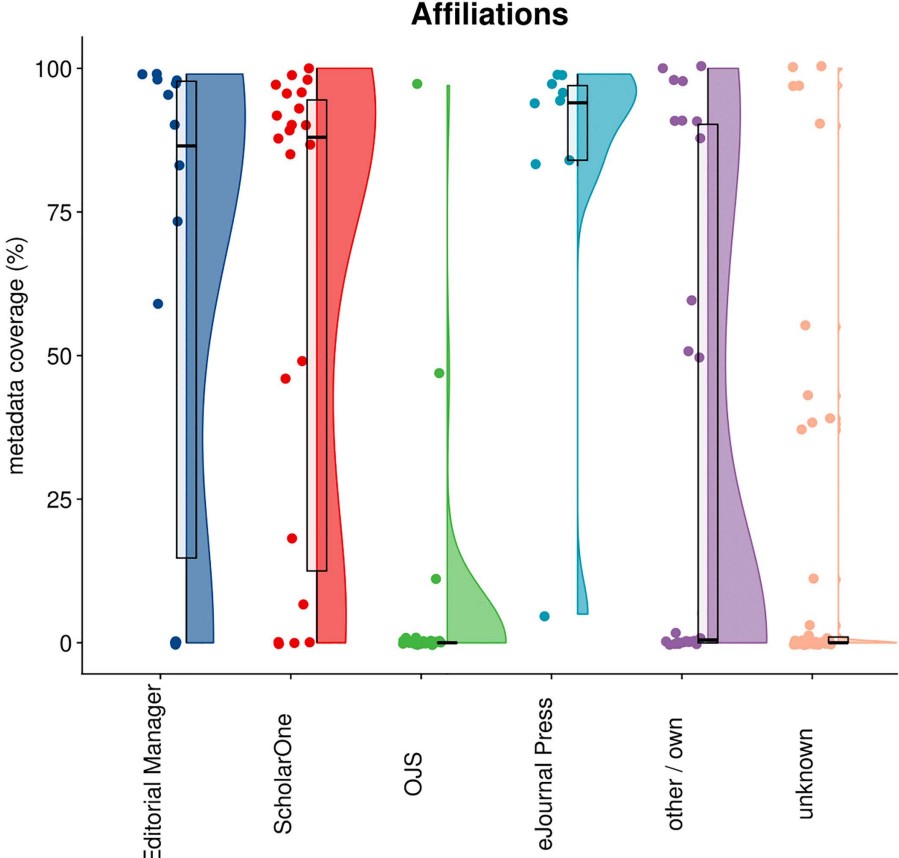

**Fig 3. Distribution of affiliation metadata coverage in Crossref by publisher, categorized by their submission system.**

An important caveat must be noted when interpreting patterns of reference metadata: unlike other metadata types examined in this study, reference information is typically not collected directly through submission systems during manuscript submission. Instead, references are extracted from manuscript files at a later stage in the production workflow. The differences we observe between submission systems are therefore most likely not explained by which submission system a publisher uses, but rather reflect downstream processes and publisher policies. A notable exception is OJS, which is a more integrated system that handles both submission, production and hosting.

Publisher policies seem to play an especially important role. While it is not mandatory to submit reference data to Crossref, it is strongly encouraged, particularly for publishers using Crossrefs Cited-by service [33]. The open citation landscape has shifted significantly in recent years, with most of the larger publishers joining the Initiative for Open Citations and Crossref's 2022 decision to make all deposited reference lists openly available [34]. The strong performance of many publishers likely reflects these developments and the growing recognition of open citations' value. However, the substantial variation between publishers also demonstrates that even when production workflows are apparently capable of extracting and depositing reference data, some publishers continue to withhold this data indicating that policy decisions—not technical capabilities—are the primary driver of reference metadata completeness..

Regarding OJS, it is known that this is a highly customizable system. Poor performance of OJS in reference metadata capture and propagation might be due to the fact that publishers or their journals have not installed the available plugins.

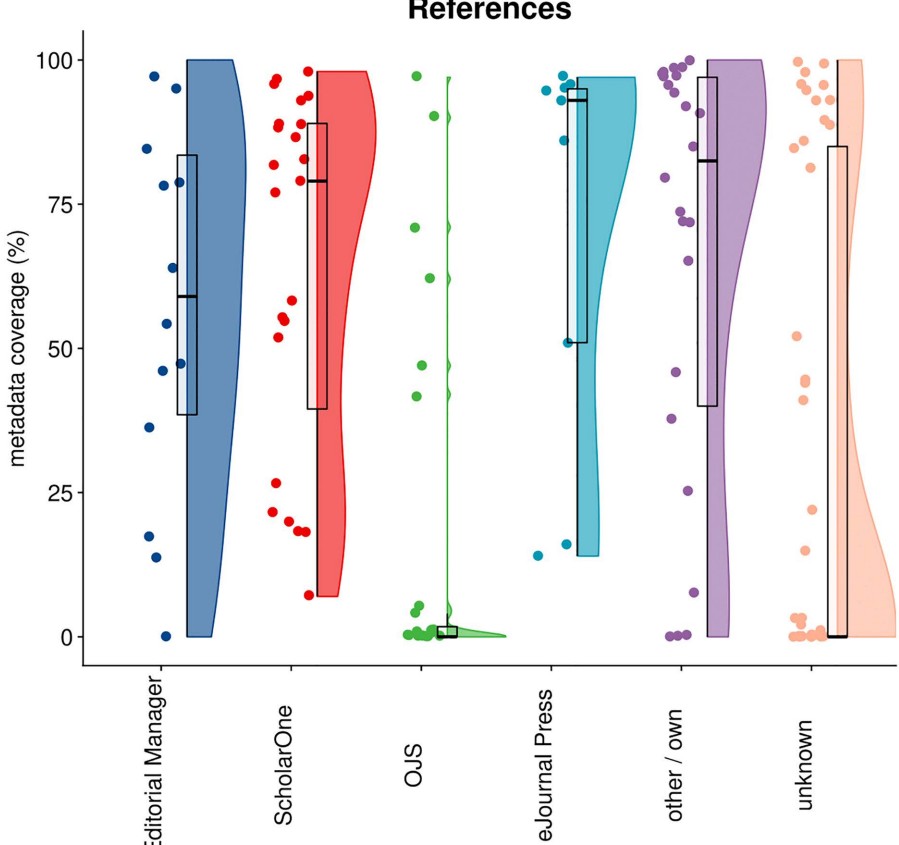

**Fig 4. Distribution of reference metadata in Crossref by publisher, categorized by their submission system.**

## 4.4 Funding information

Fig 5 displays funding metadata coverage in Crossref by publisher, categorized by submission system. Overall coverage is lower across all systems compared to other metadata types. In total, out of 11,2M articles, 23.7% had funding metadata. Both Editorial Manager and ScholarOne exhibit wide variation, with some publishers submitting no funding information at all and others reaching around 75% coverage. Open Journal Systems appears to struggle with capturing well-structured funding metadata, while eJournals Press shows relatively strong performance, with several publishers reaching 75% coverage. Other known submission systems (including in-house solutions) also show wide variation, ranging from very high coverage (over 80%) to none at all. Publishers using unknown submission systems generally perform poorly, either struggling to capture funding data or not prioritizing it.

When interpreting these results, several important caveats need to be taken into account. First, not all journal articles include funding information, since not all research outputs result from externally funded research. Second, submitting funding information to Crossref is not mandatory. The disciplinary profile of publishers also plays a role in funding meta-data completeness, as such information is more commonly acknowledged in the natural sciences than in the social sciences and humanities [35].

More importantly, it is known that publishers don't rely solely on their submission systems to gather this information [10,28]. Many collaborate with third-party providers to extract funding data from manuscript text. Some publishers have developed workflows that compare information extracted from manuscripts and through submission systems and feed this

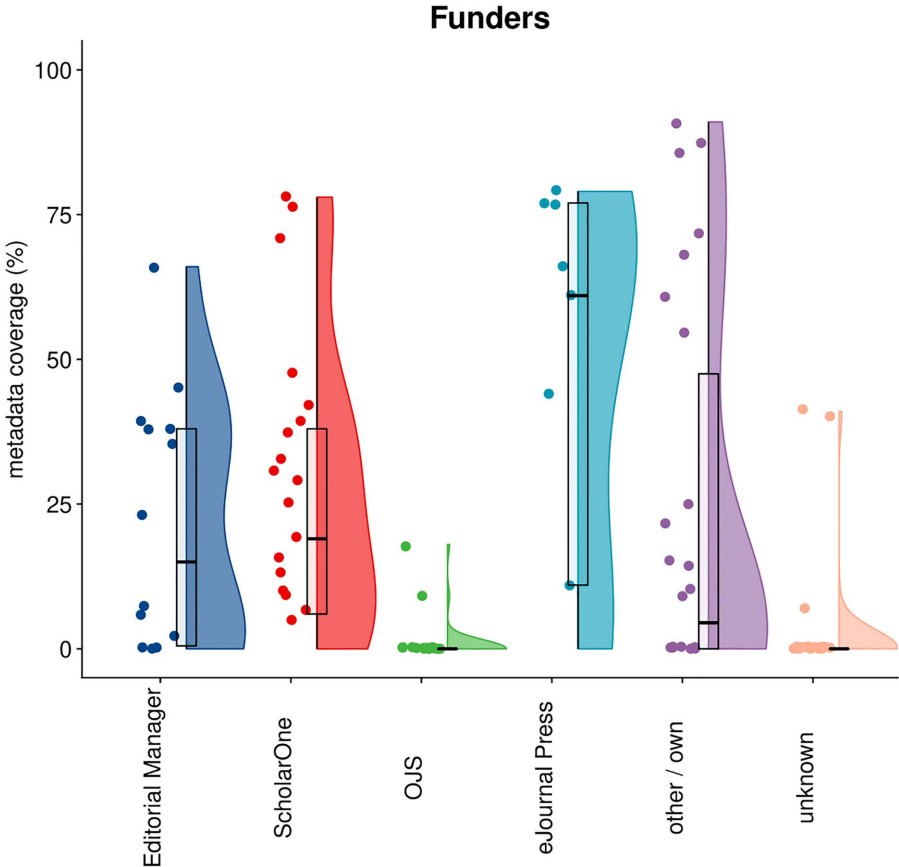

**Fig 5. Distribution of open funding metadata in Crossref by publisher, categorized by their submission system.**

back to authors, allowing them to update and correct the data. Given these varied approaches to the collection of funding metadata, we should be cautious to attribute higher percentages of funding information in Crossref to the effectiveness of particular submission systems.

Low levels of funding metadata exhibited by publishers using OJS, could be explained by the fact that OJS is a highly customizable system. Publishers or their journals may not have installed the available plugin which supports deposition of funding information [36].

## 4.5 ORCIDs

Fig 6 displays coverage of ORCiD IDs in author metadata in Crossref by publisher, categorized by submission system. Out of 11,2M articles, 40,8% had at least one ORCID included in the metadata. The patterns reveal significant variation in the level of ORCID integration across platforms. Editorial Manager and ScholarOne show broad distributions with substantial variation: while many publishers achieve high ORCID coverage (70–95%), others remain at very low levels. eJournals Press shows consistently strong performance, with most publishers clustered above 80% coverage, suggesting effective ORCID integration. Again, the performance of OJS falls behind, with the majority of publishers achieving less than 20% ORCID coverage and very few exceeding 50%. Publishers using "other/own" systems show moderate performance with wide variation, while those with "unknown" systems generally cluster at low coverage levels, similar to the pattern observed for other metadata types.

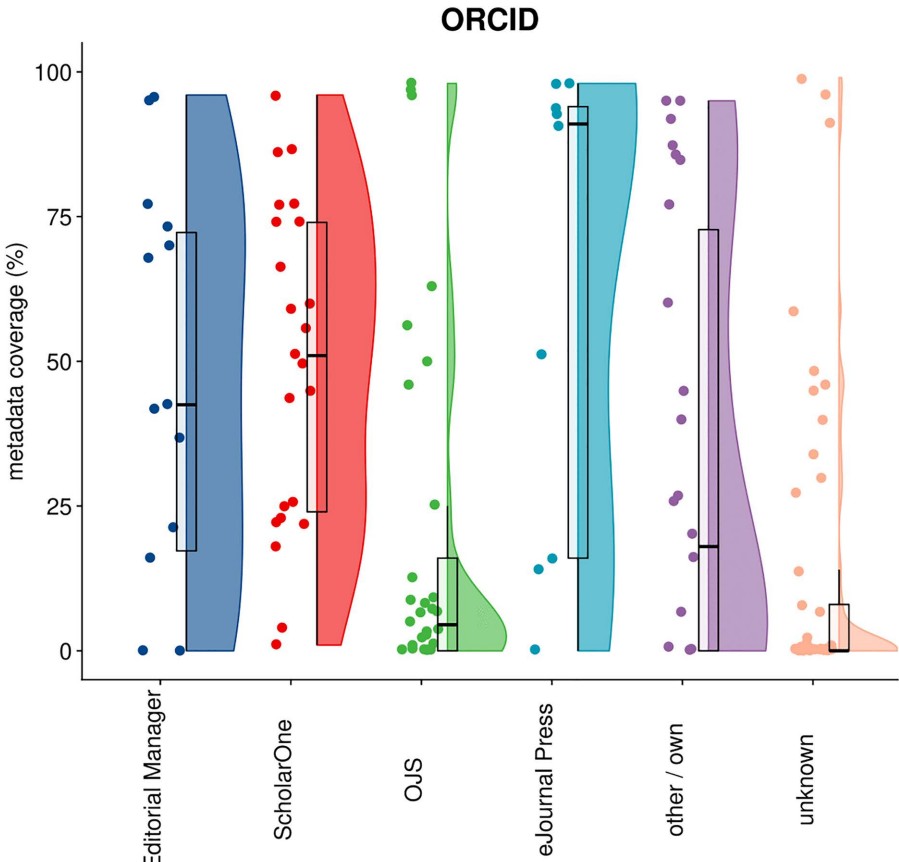

**Fig 6. Distribution of articles with at least one ORCID in Crossref by publisher, categorized by their submission system.**

The ORCID results are particularly significant given the growing emphasis on persistent author identification in scholarly publishing [37,38]. The strong performance among some Editorial Manager, ScholarOne, and eJournals Press publishers points to successful integration with ORCID authentication workflows during manuscript submission. The substantial variation in the level of ORCID registration by publishers using the same systems could be an indication that ORCID adoption depends on publisher policies. However, disciplinary and regional variations in the adoption of ORCID might also play a role [37].

### 4.6 Licenses

Fig 7 shows the distribution of license information submitted to Crossref by publishers, grouped by the submission systems they use. License information specifies the terms under which an article can be used, shared, and reused (e.g., Creative Commons licenses). Only license information for the version of record and accepted manuscript are taken into account; licenses for text and data mining are not included [6]. Out of 11,2M articles, 53.5% had license metadata. Overall, the distribution is very wide: many publishers deposit license information for all of their articles, while others do not provide this metadata at all.

This variation suggests that all submission systems are technically capable of capturing license information, and that the decision whether or not to include license information in the metadata will depend on publisher policies. An important caveat needs to be taken into account, however, when interpreting these figures. Publishers will not ask authors to specify

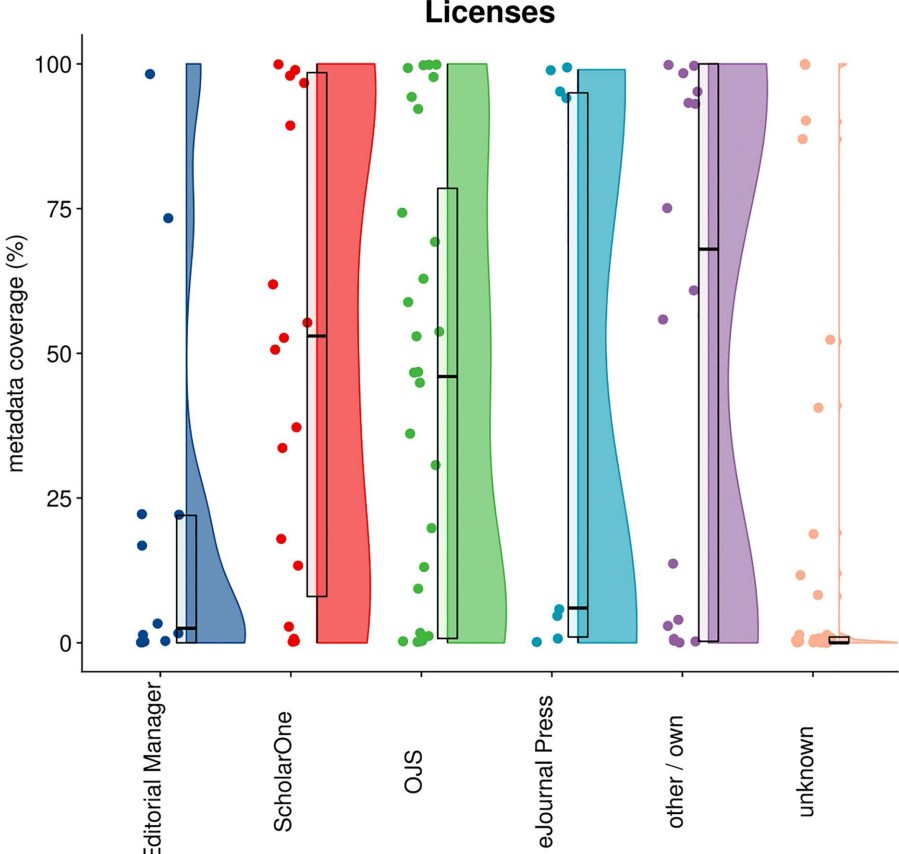

**Fig 7. Distribution of license information in Crossref by publisher, categorized by their submission system.**

their choice of license during (first) submission. Typically this will only be done after the review and acceptance of a paper. Submission systems will therefore likely have a less prominent role in collecting this metadata and consequently also on the completeness and coverage of license information in Crossref. Here again, OJS is an exception as this is a more integrated system, combining submission, production and hosting.

## 5. Concluding remarks and policy consequences

The aim of this study was to investigate the association between the level of metadata that publishers deposit with Crossref and the submission systems that they deploy for their journals. To answer this question we collected information about the submission systems used by the 153 publishers with the largest publication volume in Crossref in 2023/2024. Our analysis reveals several important patterns across a selection of six metadata types: abstracts, affiliations, references, funding information, ORCIDs, and license information.

Our main conclusion is that metadata coverage in Crossref differs widely by platform for all six metadata types examined in this study. eJournalPress and ScholarOne tend to show higher levels of coverage across most metadata types, whereas Open Journal Systems lags notably for affiliations, references, ORCIDs and funding information. Most importantly, however, we observe large differences within every system — publishers using the same submission platform show very different levels of metadata completeness, suggesting other factors may play a role rather than technical limitations of submission systems only.

As we have illustrated in Fig 1, submission systems represent a critical first step in the journal production workflow that captures metadata. However, hosting and production systems are also important in preserving this metadata and ensuring it propagates through the production process until its submission to Crossref. Publishers also work with third-party providers to extract metadata from the manuscript text, and may also decide whether to make specific metadata types available through Crossref or not. Each step in this workflow presents potential bottlenecks for the correct capture, retention and propagation of metadata from submission to deposition in Crossref.

In the context of these potential bottlenecks in publishers' production workflows, we identify four options why metadata may not find its way into Crossref:

1. Data collection gaps: publishers may not ask authors for specific metadata fields upon submission, or may not match metadata provided or extracted as free text to persistent identifiers.

2. Technical: metadata captured by a submission system is captured but lost further down the production line. This can happen when systems are not well integrated.

3. Implementation choices: while systems may possess the required functionalities, publishers may have configured their submission system such that it does not collect or propagate certain metadata types. This includes the use of specific plugins for OJS.

4. Deliberate policy: a submission system is able to collect the metadata, but the metadata is not submitted to Crossref by the publisher deliberately. Abstracts are an example where some publishers clearly collect this data but withhold them from Crossref for commercial reasons.

Our analysis shows consistently weaker performance from publishers using OJS across all metadata types. As OJS is a system widely adopted by publishers committed to open science, it seems unlikely that these publishers are opposed to making metadata openly available. The low coverage is more plausibly due to limited awareness of available plugins or insufficient technical expertise to fully utilize the system's capabilities. This points to a clear opportunity for targeted outreach and capacity-building efforts to ensure that the technical functionalities of the system are effectively leveraged. This is one of the objectives of the recently renewed partnership between Crossref and PKP [32].

Due to data availability, our analysis was focused on manuscript submission systems only. These systems are an important enabling factor for publishers in collecting metadata, but it is important to be cautious about attributing metadata completeness in Crossref only to the use of particular submission systems. In this study we have been careful not to suggest this. Future research should examine production and hosting systems' roles in metadata completeness, as well as the use of (automated) extraction of metadata from full text submissions. As this is not openly available information, more elaborate methods will be needed for that, including looking at metadata included in PDFs and examining HTML structures of published articles.

In the meantime, our results confirm that submission systems are where the lifecycle of metadata begins. Improving these systems and improving the use of the available technical functionalities by publishers seems to offer an important avenue for improving open research information via Crossref.

Here, the differences observed in our analysis might be helpful to surface best practices: publishers who do well in collecting and propagating metadata may serve as an example for publishers using the same submission systems but doing less well in this area. Hopefully this can prompt a constructive dialogue between publishers and system vendors on technical aspects, and between publishers, institutions and funders, on the importance of capturing good quality metadata and making that metadata available as open metadata, including through Crossref. For institutions, this could be an element to include as a requirement in their negotiations with publishers. Funders can take more responsibility for funding metadata by giving stricter instructions to their grantees on how to acknowledge funding and/or by registering persistent identifiers and metadata for their grants.

International collaborations, like those in the Initiative for Open Citations, the Initiative for Open Abstracts and – recently – the Barcelona Declaration on Open Research Information have proved to be powerful vehicles to bring coordination and change.

By working collectively across the research ecosystem – from system developers and publishers to institutions and funders – we can ensure that the metadata infrastructure supporting scholarly communication becomes as open, complete, and interoperable as the research it describes.

## Author contributions

**Conceptualization:** Hans de Jonge.

**Data curation:** Hans de Jonge, Bianca Kramer.

**Formal analysis:** Hans de Jonge, Bianca Kramer.

**Investigation:** Hans de Jonge, Bianca Kramer.

**Methodology:** Hans de Jonge, Bianca Kramer.

**Software:** Bianca Kramer.

**Visualization:** Bianca Kramer.

**Writing – original draft:** Hans de Jonge, Bianca Kramer.

**Writing – review & editing:** Hans de Jonge, Bianca Kramer.

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
