## [Decision Letter · Decision Letter 0]

2 Dec 2025

Dear Dr. de Jonge,

Thank you for submitting your manuscript to PLOS ONE. After careful consideration, we feel that it has merit but does not fully meet PLOS ONE’s publication criteria as it currently stands. Therefore, we invite you to submit a revised version of the manuscript that addresses the points raised during the review process.

We look forward to receiving your revised manuscript.

Kind regards,

Micah Altman

Academic Editor

PLOS ONE

Journal Requirements:

“BK is executive director of the Barcelona Declaration on Open Research Information. HdJ is co-chairing the working group on funding metadata under the Barcelona Declaration. The authors write in a personal capacity and views they share in this article do not necessarily express the opinion of their employers.”

Additional Editor Comments :

The reviewers note a number of methodological issues that should be addressed in revision. Of particular concern are revisions to ensure verifiability/reproducibility and to improve data quality ( e.g. journal vs. publisher granularity; missing data robustness)

Reviewers' comments:

Reviewer's Responses to Questions

**Comments to the Author**

1. Is the manuscript technically sound, and do the data support the conclusions?

Reviewer #1: Partly

Reviewer #2: No

2. Has the statistical analysis been performed appropriately and rigorously?

Reviewer #1: No

Reviewer #2: No

3. Have the authors made all data underlying the findings in their manuscript fully available?

Reviewer #1: No

Reviewer #2: No

4. Is the manuscript presented in an intelligible fashion and written in standard English?

Reviewer #1: Yes

Reviewer #2: Yes

Reviewer #1: This is worthwhile work that can benefit from a number of improvements, mainly to improve its soundness and verifiability. It examines an interesting and consequential research question, namely whether manuscript submission systems are correlated with the quality of metadata deposited on Crossref.

Regarding significance, the motivation behind this work is discussed in the introduction. The presented motivation could be improved by circling back to submission systems: What actions could publishers and researchers take in view of this research? The presented contributions can clearly impact scientific publishing practice and bibliometrics as described in the concluding sections.

Regarding novelty, the work's contributions are original with respect to the state-of-the-art. Existing work in the area is coherently analyzed and synthesized, and the study's contributions are clearly identified in its context in the first two sections

Regarding soundness, the study's contributions address its research questions, but in some cases the applied research methods lack the required rigor. The study's accuracy suffers from the decision to examine submission systems at the level of publishers, rather than journals. As the authors admit in p. 9, publishers often operate several submission systems in parallel. The study could be easily improved by tracking individual journals and employing the publisher only as a shortcut when the publisher operates only a single system. Furthermore, the study is based on a non-peer reviewed data set. Given this, one would expect the provision of additional details regarding the data set and the employment of several safeguards regarding its veracity. In addition to presenting rainclouds, the authors could form research hypotheses and examine the reported correlations with statistical methods. Finally, the quality of the provided metadata (e.g. in terms of DOIs and RORs) is not examined. The contribution's evaluation can be further improved by triangulating the findings with other sources, though I don't think that this is required for the study's publication. Threats to validity and their effect on the results' soundness are not explicitly presented, though they are hinted throughout the manuscript.

Regarding verifiability, the method's description and the provided data allow the independent verification of the outlined findings. The study's verifiability would benefit by providing additional materials to support independent verification or replication of the paper’s claimed contributions especially regarding the methods that were employed for the analysis and the survey's responses.

Regarding presentation, the paper is well structured. The writing is clear and easy to understand. The text is free from typos and style errors. The formatting follows the provided instructions. The abstract is self-standing and summarizes succinctly the paper's essence. The provided figures help the reader understand the presented material. Their design can be improved in terms of aesthetics and readability by rendering them in a vector format to ensure high-quality presentation. More importantly, the authors should clarify the meaning of dots in the rainclouds. They appear to be too few to represent articles and too many to represent publishers or journals.

Below is a list of more detailed comments.

Page 1, line 19: Please consider removing “Of course”

Page 2, line 13: Please describe what these initiatives are about.

Page 2, line 17: Please clarify “it”.

Page 4, line 14: Please provide a citation backing the claim.

Page 10, line 14: Production systems could be examined through metadata embedded in the published PDFs. Similarly, hosting systems could be discernable through HTTP headers and URLs. I'm not suggesting this as an action point for this study, but as an idea for a future one.

Figure 1: Consider changing “adjust the text to form complete sentences ” into “ phrases. Also provide the figure in vector format to ensure high-quality presentation.”.

Page 12, line 2: Please clarify the level of normalization expected here, e.g. ROR, DOI. This should then be reflected in the presented results.

Section 4.1: Consider noting that the data distribution in the unknown category resembles the union of the others.

Section 4.1: Please provide descriptive statistics regarding the analyzed data. How many Crossref records were analyzed? What was their distribution among publishers and journals? What were the aggregate figures regarding metadata availability?

Section 4.6: Please clarify the meaning of licensing information and how it is related to submission. How common is it for authors to have a say in it?

Reviewer #2: This paper takes 153 publishers as examples and explores the correlation between the submission system and the metadata level of Crossref through questionnaire surveys and statistical analysis. However, there are still some unclear points in the metadata selection, method design, and data analysis of this paper.

1. This paper selects six types of metadata based on reference [6] for research. However, there is no relevant explanation in the research background and method section as to why these six types were chosen. Since there are many types of metadata, the criteria for determining which metadata are important and which are not are not explained in this paper, nor are the existing research results listed. This may make the research significance of this paper unclear and affect the rationality of the method design.

2. This paper mentions factors that may be related to Crossref metadata, including submission systems, hosting and production systems, third-party suppliers, and specific policies of publishers. And it is also mentioned that the information obtained from the questionnaire survey was not ideal. There is one point that is not clearly explained: whether the publishers were indeed submitting metadata to Crossref through the submission system. If so, the subsequent research on the correlation between the two would be logical. If not, it would be illogical. Therefore, it is suggested that the results of the survey obtained in the method section be presented more clearly.

3. In terms of data processing and analysis, in addition to the above-mentioned issue of why six types of metadata were chosen, some data in the paper seem inconsistent. In some places, 153 publishers are mentioned, while in others, 154 publishers are mentioned. Also, the types of metadata should be six, but in the conclusion, there are five, seven, sixteen, etc. Can it be clearly stated what the metadata are?

**Do you want your identity to be public for this peer review?** For information about this choice, including consent withdrawal, please see our Privacy Policy

Reviewer #1: **Yes:** Diomidis Spinellis

Reviewer #2: No

---

## [Author Response · Author response to Decision Letter 1]

4 Jan 2026

Dr. Micah Altman

Academic Editor

PLOS ONE

Leiden / Houten, January 5 2026

Dear Dr. Altman,

We are pleased to resubmit our revised manuscript entitled "Manuscript submission systems and metadata completeness in Crossref: patterns and associations" [PONE-D-25-53920] for consideration in PLOS ONE. We sincerely thank the reviewers for their thoughtful and constructive comments, which have substantially improved the quality and clarity of our work. We are also grateful to you for the opportunity to revise and resubmit our manuscript. In the following pages, we provide detailed point-by-point responses to each of the reviewers' comments, explaining how we have addressed their concerns in the revised manuscript.

Style requirements

We have updated our manuscript to ensure it follows the style requirements of PLOSOne.

Participants consent

Our study consisted of two components: (1) a quantitative analysis of publicly available metadata from Crossref, and (2) a survey among 153 publishers. For the Crossref metadata analysis, no consent was required as all data are publicly accessible as part of the earlier work by Van Eck and Waltman.

For the publisher survey, respondents were informed at the beginning of the survey that their responses would be used for this study. Participation was voluntary, and by completing and submitting the survey, respondents provided implied consent for their data to be used in this research. However, we did not explicitly request permission to share verbatim responses to open-ended questions. Therefore, in our manuscript and accompanying dataset, we only report factual information from the survey responses (which submission system the publisher is using), without including direct quotes or identifiable information from open-ended responses. We have included an ethics statement in the Methods section to reflect this information although we are not sure this is required for this type of research.

Competing interests

We have updated our competing interests statement as prescribed:

BK is executive director of the Barcelona Declaration on Open Research Information. HdJ is co-chairing the working group on funding metadata under the Barcelona Declaration. The authors write in a personal capacity and views they share in this article do not necessarily express the opinion of their employers. *This does not alter our adherence to PLOS ONE policies on sharing data and materials*.

Point by point response to the reviewers comments:

Reviewer #1: This is worthwhile work that can benefit from a number of improvements, mainly to improve its soundness and verifiability. It examines an interesting and consequential research question, namely whether manuscript submission systems are correlated with the quality of metadata deposited on Crossref.

Regarding significance, the motivation behind this work is discussed in the introduction. The presented motivation could be improved by circling back to submission systems: What actions could publishers and researchers take in view of this research?

Response:

We agree with the reviewer and have added a new paragraph to the introduction where we highlight that our research can have important consequences for evidence informed actions publishers and submission system providers can take based on our research. The reviewers might be interested to learn that we have already been contacted by publishers based on the findings presented in our preprint.

The presented contributions can clearly impact scientific publishing practice and bibliometrics as described in the concluding sections.

Regarding novelty, the work's contributions are original with respect to the state-of-the-art. Existing work in the area is coherently analyzed and synthesized, and the study's contributions are clearly identified in its context in the first two sections

Regarding soundness, the study's contributions address its research questions, but in some cases the applied research methods lack the required rigor. The study's accuracy suffers from the decision to examine submission systems at the level of publishers, rather than journals. As the authors admit in p. 9, publishers often operate several submission systems in parallel. The study could be easily improved by tracking individual journals and employing the publisher only as a shortcut when the publisher operates only a single system.

Response:

We appreciate the reviewer's suggestion to conduct the analysis at the journal level. However, we respectfully disagree that this represents a methodological weakness or that such an improvement would be "easy" to implement.

First, our research question explicitly focuses on publishers and their role in making metadata openly available through Crossref. Publishers, not individual journals, make the organizational and policy decisions about metadata deposition, negotiate with system vendors, configure their technical infrastructure, and ultimately bear responsibility for metadata completeness in Crossref. Shifting the analysis to the journal level would detract from this focus on publisher responsibilities and obscure the organizational factors that drive metadata practices. Moreover, our core finding - that metadata completeness varies substantially even among publishers using the same submission system - demonstrates that organizational policies and practices, not just technical infrastructure, drive metadata quality. Journal-level analysis would add granularity without fundamentally altering this insight.

Second, we disagree with the reviewer about the practical feasibility of the suggested journal-level approach. Approximately 140 of our 153 publishers use a single submission system. For these, journal-level analysis would yield identical results. The remaining 10-20 largest publishers operating multiple systems collectively manage approximately 15,000 journal titles. Since submission system data at the journal level is not publicly available and must be collected manually (as we did for publishers), this would require a 100-fold increase in verification work - far from an "easy improvement", and well beyond our resources.

We therefore maintain that our publisher-level analysis is both methodologically appropriate for examining publisher responsibilities in metadata deposition and provides robust evidence for our conclusions. The suggested alternative would require disproportionate resources for marginal analytical benefit while shifting focus away from the organizational accountability that is central to our research question.

Furthermore, the study is based on a non-peer reviewed data set. Given this, one would expect the provision of additional details regarding the data set and the employment of several safeguards regarding its veracity.

Response:

We contacted the authors of the original dataset (Van Eck and Waltman) to verify its peer review status. Their work has undergone formal peer review through two stages. It was first presented at the 2021 International Conference on Scientometrics & Informetrics, where it received reviews from two independent reviewers. Subsequently, the work was posted as a preprint with the underlying dataset and submitted for open peer review on the PeerRef platform. Following the platform's closure, the two positive reviews are no longer publicly accessible, though their existence can be verified through the DOIs 10.58505/XAGF4544 and 10.58505/CWBO6951. The credibility of both the work and its underlying dataset is further supported by the numerous references in other publications: (https://scholar.google.com/scholar?cites=492535894070598693&as_sdt=2005&sciodt=0,5&hl=nl).

While the dataset cannot be directly reproduced without access to both the exact Crossref data snapshot and the accompanying analysis scripts used by Van Eck and Waltman, the percentages metadata coverage per publisher and publication type can be compared to and verified against those in the Crossref Participation Reports (https://www.crossref.org/members/prep/), with the understanding that these cover publications from 2024-2026 compared to 2023-2024 in the original dataset.

In addition to presenting rainclouds, the authors could form research hypotheses and examine the reported correlations with statistical methods.

Response:

When we started this study, we initially intended to describe the relationship between metadata completeness and submission systems in statistical terms. However, we ultimately moved away from this approach and adopted a more descriptive method. As we stress in our paper, the relationship between submission system and completeness in Crossref is not straightforward: publishers sometimes use multiple systems in parallel, but above all: the chain of systems involved in the production process (from submission to production and hosting) is complex. Metadata can be lost at all stages in that process and therefore it is important to be cautious in our conclusions and avoid claiming direct one-to-one statistical correlations between submission systems and metadata completeness in Crossref. To reflect this, we have replaced the term 'correlation' with 'relationship' throughout our manuscript.

Finally, the quality of the provided metadata (e.g. in terms of DOIs and RORs) is not examined.

Response:

Our study does indeed not attempt to assess metadata quality in Crossref; rather, it focuses exclusively on the presence or absence of six metadata types, as we acknowledge on page 8. This scope is determined by our source dataset (Van Eck and Waltman), which similarly examines metadata coverage in terms of presence/absence rather than quality. Assessing metadata quality would require a substantially more in-depth study beyond the scope of this article, which focuses specifically on the coverage of metadata as provided by publishers.

The contribution's evaluation can be further improved by triangulating the findings with other sources, though I don't think that this is required for the study's publication.

Response:

We acknowledge that this would be an interesting exercise for future study, but not for the specific question that is central to this article. In this study, we limited ourselves to Crossref because this is the data that is supplied directly by publishers. Of course, additional metadata can be found in other databases. However, comparing these sources introduces confounding factors, namely where the information has been collected through other means (e.g., harvesting landing pages, repositories, etc.). This makes it impossible to establish a relationship between submission systems operated by publishers, which is the primary objective of our study.

Threats to validity and their effect on the results' soundness are not explicitly presented, though they are hinted throughout the manuscript.

Response:

We have added a new section 3.5 (Limitations) that explicitly consolidates the various threats to validity that were previously scattered throughout the manuscript. This section now clearly presents the scope and constraints of our study, including our focus on submission systems only, the role of publisher policies and third-party services, and limitations in our data collection approach.

Regarding verifiability, the method's description and the provided data allow the independent verification of the outlined findings. The study's verifiability would benefit by providing additional materials to support independent verification or replication of the paper’s claimed contributions especially regarding the methods that were employed for the analysis and the survey's responses.

Response:

We appreciate the reviewer's comments on the verifiability and replicability of our study. We believe we have shared all data necessary for the replication of this study's findings. Specifically, we have made available:

- The aggregated metadata coverage data from Crossref for 153 publishers across six metadata elements (reused from Van Eck & Waltman);

- Data about the manuscript submission systems these publishers use, as collected by the authors through systematic website examination. For verifiability, we have updated our data set to include URL’s to the publishers' websites as documentation;

- Where available, validation of this information from our publisher survey responses.

- To strengthen the replicability of our work we have included the R scripts of the raincloud plots. These materials, combined with the publicly accessible Crossref metadata (available through the Crossref API), enable independent verification and replication of our findings.

With regards to the veracity of the dataset we reuse, we refer to our earlier response above.

Regarding presentation, the paper is well structured. The writing is clear and easy to understand. The text is free from typos and style errors. The formatting follows the provided instructions. The abstract is self-standing and summarizes succinctly the paper's essence. The provided figures help the reader understand the presented material. Their design can be improved in terms of aesthetics and readability by rendering them in a vector format to ensure high-quality presentation.

Response:

A vector format for the figures is provided to the journal separately.

More importantly, the authors should clarify the meaning of dots in the rainclouds. They appear to be too few to represent articles and too many to represent publishers or journals.

Done

Below is a list of more detailed comments.

Page 1, line 19: Please consider removing “Of course”

Done

Page 2, line 13: Please describe what these initiatives are about.

Done

Page 2, line 17: Please clarify “it”.

Done

Page 4, line 14: Please provide a citation backing the claim.

Done

Page 10, line 14: Production systems could be examined through metadata embedded in the published PDFs. Similarly, hosting systems could be discernable through HTTP headers and URLs. I'm not suggesting this as an action point for this study, but as an idea for a future one.

Response:

We thank the reviewer for this interesting suggestion and will certainly look into this for future study. We have updated our manuscript to reflect that.

Figure 1: Consider changing “adjust the text to form complete sentences ” into “ phrases. Also provide the figure in vector format to ensure high-quality presentation.”.

Response:

We thank the reviewer for this suggestion. However, we believe that expanding the labels to complete sentences or longer phrases would compromise the figure's readability. We will provide Figure 1 in vector format to ensure high-quality presentation.

Page 12, line 2: Please clarify the level of normalization expected here, e.g. ROR, DOI. This should then be reflected in the presented results.

Response:

Crossref data are well-structured. As a registration agency of the International DOI Foundation, Crossref follows the ISO/IEC 11179 Metadata Registry (MDR) standard, which specifies a schema for recording both the meaning and technical structure of the data for unambiguous usage (https://www.crossref.org/documentation/schema-library/).

Since Van Eck and Waltman used data directly obtained from Crossref, there is no issue with normalization of DOIs. As described in section 3.3 of our study, for each metadata type, Van Eck and Waltman counted presence of information in the respective metadata field as a positive. No further attempt was made to analyse and check the information provided in the metadata fields, e.g. length and textual characteristics of the abstracts, completeness of affiliation information (including presence or absence of ROR), number and completeness of references (including presence or absence of DOIs), type of funding information (funder name and/or ID, award number), authentication of ORCIDs and standardization of license information.

Thus, normalization of information in the respective metadata fields (most relevant for string-based fields) are not relevant for the presentation and interpretation of results in our study. The results of van Eck and Waltman, and by extension our study, provide an initial insight into the coverage of metadata (number of records with that metadata type present), rather than their qual

---

## [Decision Letter · Decision Letter 1]

4 Feb 2026

Dear Dr. de Jonge,

Thank you for submitting your manuscript to PLOS ONE. After careful consideration, we feel that it has merit but does not fully meet PLOS ONE’s publication criteria as it currently stands. Therefore, we invite you to submit a revised version of the manuscript that addresses the points raised during the review process.

plosone@plos.org . A letter that responds to each point raised by the academic editor and reviewer(s). You should upload this letter as a separate file labeled 'Response to Reviewers'.A marked-up copy of your manuscript that highlights changes made to the original version. You should upload this as a separate file labeled 'Revised Manuscript with Track Changes'.An unmarked version of your revised paper without tracked changes. You should upload this as a separate file labeled 'Manuscript'.

We look forward to receiving your revised manuscript.

Kind regards,

Micah Altman

Academic Editor

PLOS One

Journal Requirements:

Additional Editor Comments:

Please consider this article as accepted, with the opportunity for minor revision of the text based on Reviewer 1 comments.

Reviewers' comments:

Reviewer's Responses to Questions

**Comments to the Author**

Reviewer #1: All comments have been addressed

Reviewer #2: (No Response)

2. Is the manuscript technically sound, and do the data support the conclusions?

Reviewer #1: Yes

Reviewer #2: (No Response)

3. Has the statistical analysis been performed appropriately and rigorously?

Reviewer #1: N/A

Reviewer #2: (No Response)

4. Have the authors made all data underlying the findings in their manuscript fully available?

Reviewer #1: Yes

Reviewer #2: (No Response)

5. Is the manuscript presented in an intelligible fashion and written in standard English?

Reviewer #1: Yes

Reviewer #2: (No Response)

Reviewer #1: The authors have covered most issues identified in the earlier review and have either explained their choices' rationale or they have adjusted the submitted manuscript.

I suggest that the authors detail (where needed) their review responses in the paper as rationale for their methodological decisions.

I appreciate that the several potential sources of bias or error are identified in the new “3.5 Limitations” section. Kindly verify that these include all the points raised by the reviewers.

It would be helpful to identify the number to journals associated with publishers operating a single submission system versus the ones associated with publishers operating multiple submission systems.

Reviewer #2: (No Response)

**Do you want your identity to be public for this peer review?** For information about this choice, including consent withdrawal, please see our Privacy Policy

Reviewer #1: No

Reviewer #2: No

---

## [Author Response · Author response to Decision Letter 2]

24 Feb 2026

Response to Reviewers

We once again thank the reviewers for their thoughtful comments. We are happy that our revisions have been able to satisfy the reviewers. Below is a response to the remaining points that have been raised:

Reviewer #1: The authors have covered most issues identified in the earlier review and have either explained their choices' rationale or they have adjusted the submitted manuscript.

I suggest that the authors detail (where needed) their review responses in the paper as rationale for their methodological decisions.

We have gone through our rebuttal letter and concluded that all our choices’ rationale are covered in the manuscript. In two instances we found that the formulation in our rebuttal was more clear than our original formulation. We have updated the text accordingly.

I appreciate that the several potential sources of bias or error are identified in the new “3.5 Limitations” section. Kindly verify that these include all the points raised by the reviewers.

We have gone through the reviews and can confidently say all points raised our now covered in the new limitations section.

It would be helpful to identify the number to journals associated with publishers operating a single submission system versus the ones associated with publishers operating multiple submission systems.

We have included these on page 6

---

## [Editor Report · Decision Letter 2]

6 Mar 2026

Manuscript submission systems and metadata completeness in Crossref: patterns and associations

PONE-D-25-53920R2

Dear Dr. de Jonge,

We’re pleased to inform you that your manuscript has been judged scientifically suitable for publication and will be formally accepted for publication once it meets all outstanding technical requirements.

Kind regards,

Micah Altman

Academic Editor

PLOS One
---

## [Editor Report · Acceptance letter]

PONE-D-25-53920R2

PLOS One

Dear Dr. de Jonge,

I'm pleased to inform you that your manuscript has been deemed suitable for publication in PLOS One. Congratulations! Your manuscript is now being handed over to our production team.

Kind regards,

on behalf of

Dr. Micah Altman

Academic Editor

PLOS One